# Instantaneous and Non-Instantaneous Impulsive Boundary Value Problem Involving the Generalized $\psi$-Caputo Fractional Derivative

**Dongping Li [1], Yankai Li [2,\*], Fangqi Chen [3] and Xiaozhou Feng [1]**

[1]  School of Sciences, Xi'an Technological University, Xi'an 710021, China
[2]  School of Automation and Information Engineering, Xi'an University of Technology, Xi'an 710048, China
[3]  Department of Mathematics, Nanjing University of Aeronautics and Astronautics, Nanjing 211106, China
\*  Correspondence: liyankai@xaut.edu.cn

**Abstract:** This paper studies a new class of instantaneous and non-instantaneous impulsive boundary value problem involving the generalized $\psi$-Caputo fractional derivative with a weight. Depending on critical point theorems and some properties of $\psi$-Caputo-type fractional integration and differentiation, the variational construction and multiplicity result of solutions are established.

**Keywords:** instantaneous impulse; non-instantaneous impulse; $\psi$-Caputo fractional operator; variational construction

**MSC:** 26A33; 34B15; 34A08

## 1. Introduction

Fractional calculus is an expansion of Newton Leibniz's integer order differential and integral. In recent decades, a large number of definitions of fractional calculus operators are generated with practical problem modeling requirements, such as the well known Riemann-Liouville, Caputo, Erdelyi-Kober, and Hadamard versions [1–3], and those forms play important roles in various interdisciplinary disciplines, like viscoelastic mechanics, anomalous diffusion, control theory, bioengineering, etc. [4–6]. However, many scholars discovered that some existing fractional operators may not well to describe many phenomena in the real world. Hence, a whole newly general definition is proposed recently, so-called $\psi$-Caputo-type fractional operator [7–9], which could combine the maximum number of definitions of fractional derivatives to a single one by depending upon a nonsingular kernel. The kernel function can provide free arguments to better calibrate a system [10–12]. Taking all these into account, we think that it is a promising topic for further investigation to study fractional differential equations (FDEs for short) with the generalized $\psi$-Caputo-type fractional operator.

Furthermore, the impulsive FDE can reflect the phenomenon that the state of a thing changes suddenly after being disturbed instantaneously, which is an effective means to depict the changing laws of objects. According to the duration of the change process, the impulse can be divided into the instantaneous (the definition of classical one) and non-instantaneous impulses. Most of the research on FDEs with instantaneous impulse are studied [13–15]. In 2013, Hernádez and O'Regan first proposed the non-instantaneous impulse concept based on pharmacokinetics [16], which refers to the behavior that the state is disturbed at a certain time and produces sudden changes, and it maintains the active state for a limited time interval. This work showed that the non-instantaneous impulse has more advantages in describing the human body's absorption, diffusion, and metabolism of drugs. Since then, non-instantaneous impulsive FDEs received great attention [17–20]. In [18], depending on the Weierstrass theorem, the existence of solutions was obtained for a class

of instantaneous and non-instantaneous impulsive fractional Dirichlet boundary value problems with perturbation. In view of the well known three critical points theorem due to B. Ricceri, the existence of at least three solutions for the non-instantaneous impulsive FDE was obtained in [19]. Because of the late development of non-instantaneous impulse comparing with the instantaneous impulse, many theoretical results need to be enriched and improved, so it has great potential research space and theoretical significance.

Motivated by above works, in this paper, we are concerned with a new class of instantaneous and non-instantaneous impulsive FDEs involving a $\psi$-Caputo fractional derivative

$$
\begin{cases}
{}^{C}D_{T^-}^{\alpha,\psi}({}^{C}D_{0^+}^{\alpha,\psi}x(t)) = \lambda f_i(t, x(t)), \ t \in (s_i, t_{i+1}], \ i = 0, 1, \ldots, n, \\
\Delta({}^{C}D_{T^-}^{\alpha,\psi}(I_{0^+}^{1-\alpha,\psi}x))(t_i) = I_i(x(t_i)), \ i = 1, 2, \ldots, n, \\
{}^{C}D_{T^-}^{\alpha,\psi}(I_{0^+}^{1-\alpha,\psi}x)(t) = {}^{C}D_{T^-}^{\alpha,\psi}(I_{0^+}^{1-\alpha,\psi}x)(t_i^+), \ t \in (t_i, s_i], \ i = 1, 2, \ldots, n, \\
{}^{C}D_{T^-}^{\alpha,\psi}(I_{0^+}^{1-\alpha,\psi}x)(s_i^-) = {}^{C}D_{T^-}^{\alpha,\psi}(I_{0^+}^{1-\alpha,\psi}x)(s_i^+), \ i = 1, 2, \ldots, n, \\
x(0) = x(T) = 0,
\end{cases}
\tag{1}
$$

where $\lambda > 0, 0 < \alpha \leq 1$, ${}^{C}D_{T^-}^{\alpha,\psi}$ and ${}^{C}D_{0^+}^{\alpha,\psi}$ denote the right and left $\psi$-Caputo fractional derivatives, $I_{0^+}^{1-\alpha,\psi}$ is the left $\psi$-Riemann-Liouville type fractional integral with order $1 - \alpha$. $\psi(t) \in C^1[0, T]$ is an increasing function with $\psi'(t) \neq 0$ for all $t \in [0, T]$. $I_i \in C(\mathbb{R}, \mathbb{R})$, $f_i \in C((s_i, t_{i+1}] \times \mathbb{R}, \mathbb{R})$, $0 = s_0 < t_1 < s_1 < \cdots < s_n < t_{n+1} = T$, the instantaneous impulse begins suddenly at the point $t_i$, and the non-instantaneous impulse continues during a finite interval $(t_i, s_i]$,

$$
\Delta({}^{C}D_{T^-}^{\alpha,\psi}(I_{0^+}^{1-\alpha,\psi}x))(t_i) = {}^{C}D_{T^-}^{\alpha,\psi}(I_{0^+}^{1-\alpha,\psi}x)(t_i^+) - {}^{C}D_{T^-}^{\alpha,\psi}(I_{0^+}^{1-\alpha,\psi}x)(t_i^-),
$$

$$
{}^{C}D_{T^-}^{\alpha,\psi}(I_{0^+}^{1-\alpha,\psi}x)(t_i^+) = \lim_{t \to t_i^+} {}^{C}D_{T^-}^{\alpha,\psi}(I_{0^+}^{1-\alpha,\psi}x)(t),
$$

$$
{}^{C}D_{T^-}^{\alpha,\psi}(I_{0^+}^{1-\alpha,\psi}x)(t_i^-) = \lim_{t \to t_i^-} {}^{C}D_{T^-}^{\alpha,\psi}(I_{0^+}^{1-\alpha,\psi}x)(t).
$$

It is a new issue that has not been touched yet. Some existing results, which focus on the classical fractional operators, such as [19,21,22], are improved and supplemented by choosing special kernel functions in the derivative.

## 2. Fractional Integrals and Derivatives

This section introduces some essential definitions of fractional integrals and derivatives, as well as relevant lemmas and theorems, whose involvements assist us to establish variational construction and multiplicity results for impulsive FDE (1) successfully.

We deal mainly with the $\psi$-Riemann-Liouville and $\psi$-Caputo fractional integrals and derivatives in this paper, and the reader can refer to Res. [7–9] for more information. Let $\alpha > 0, -\infty \leq a < b \leq +\infty$, $f(t)$ is an integrable function and $\psi(t) \in C^1[0, T]$ is an increasing function, with $\psi'(t) \neq 0$ for all $t \in [a, b]$. The left $\psi$-Riemann-Liouville type fractional integral and derivative of a function $f$ with respect to another function $\psi$ are, respectively, defined as:

$$
I_{a^+}^{\alpha,\psi}f(t) = \frac{1}{\Gamma(\alpha)} \int_a^t \psi'(\xi)(\psi(t) - \psi(\xi))^{\alpha-1} f(\xi)d\xi,
\tag{2}
$$

$$
D_{a^+}^{\alpha,\psi}f(t) = \left(\frac{1}{\psi'(t)}\frac{d}{dt}\right)^n I_{a^+}^{n-\alpha,\psi}f(t) = \frac{1}{\Gamma(n-\alpha)}\left(\frac{1}{\psi'(t)}\frac{d}{dt}\right)^n \int_a^t \psi'(\xi)(\psi(t) - \psi(\xi))^{n-\alpha-1}f(\xi)d\xi,
$$

where $n = [\alpha] + 1$ for $\alpha \notin \mathbb{N}$, $n = \alpha$ for $\alpha \in \mathbb{N}$.

Similar definitions can be given for the right $\psi$-Riemann-Liouville fractional integral and derivative:

$$I_{b-}^{\alpha,\psi}f(t) = \frac{1}{\Gamma(\alpha)} \int_t^b \psi'(\xi)(\psi(\xi) - \psi(t))^{\alpha-1}f(\xi)d\xi, \tag{3}$$

$$D_{b-}^{\alpha,\psi}f(t) = \left(\frac{-1}{\psi'(t)}\frac{d}{dt}\right)^n I_{b-}^{n-\alpha,\psi}f(t) = \frac{1}{\Gamma(n-\alpha)}\left(-\frac{1}{\psi'(t)}\frac{d}{dt}\right)^n \int_t^b \psi'(\xi)(\psi(\xi) - \psi(t))^{n-\alpha-1}f(\xi)d\xi. \tag{}$$

In particular, if $0 < \alpha < 1$, one has:

$$D_{a+}^{\alpha,\psi}f(t) = \left(\frac{1}{\psi'(t)}\frac{d}{dt}\right)I_{a+}^{1-\alpha,\psi}f(t) = \frac{1}{\Gamma(1-\alpha)}\left(\frac{1}{\psi'(t)}\frac{d}{dt}\right)\int_a^t \psi'(\xi)(\psi(t) - \psi(\xi))^{-\alpha}f(\xi)d\xi, \tag{4}$$

$$D_{b-}^{\alpha,\psi}f(t) = \left(\frac{-1}{\psi'(t)}\frac{d}{dt}\right)I_{b-}^{1-\alpha,\psi}f(t) = \frac{1}{\Gamma(1-\alpha)}\left(\frac{-1}{\psi'(t)}\frac{d}{dt}\right)\int_t^b \psi'(\xi)(\psi(\xi) - \psi(t))^{-\alpha}f(\xi)d\xi. \tag{5}$$

It is worth noting that, if we choose the kernel $\psi(t) = \ln t$ or $\psi(t) = t$, the $\psi$-Riemann-Liouville fractional integral and derivative can reduce into the well known Hadamard type or Riemann-Liouville type fractional integral and derivative.

**Definition 1** ([9]). *Let* $n \in \mathbb{N}$, $-\infty \le a < b \le +\infty$, $\alpha > 0$, $f(t), \psi(t) \in C^1[0, T]$ *are two functions, such that* $\psi(t)$ *is an increasing function with* $\psi'(t) \ne 0$ *for all* $t \in [a, b]$. *Then, the left and right* $\psi$-Caputo *type fractional derivatives of* $f$ *with respect to another function* $\psi$ *are, respectively, defined as:*

$$^C D_{a+}^{\alpha,\psi}f(t) = I_{a+}^{n-\alpha,\psi}\left(\frac{1}{\psi'(t)}\frac{d}{dt}\right)^n f(t) = \frac{1}{\Gamma(n-\alpha)}\int_a^t \psi'(\xi)(\psi(t) - \psi(\xi))^{n-\alpha-1}\left(\frac{1}{\psi'(\xi)}\frac{d}{d\xi}\right)^n f(\xi)d\xi,$$

$$^C D_{b-}^{\alpha,\psi}f(t) = I_{b-}^{n-\alpha,\psi}\left(-\frac{1}{\psi'(t)}\frac{d}{dt}\right)^n f(t) = \frac{(-1)^n}{\Gamma(n-\alpha)}\int_t^b \psi'(\xi)(\psi(\xi) - \psi(t))^{n-\alpha-1}\left(\frac{1}{\psi'(\xi)}\frac{d}{d\xi}\right)^n f(\xi)d\xi.$$

*In particular, if* $0 < \alpha < 1$, *one has:*

$$^C D_{a+}^{\alpha,\psi}f(t) = I_{a+}^{1-\alpha,\psi}\left(\frac{1}{\psi'(t)}\frac{d}{dt}\right)f(t) = \frac{1}{\Gamma(1-\alpha)}\int_a^t (\psi(t) - \psi(\xi))^{-\alpha}f'(\xi)d\xi, \tag{6}$$

$$^C D_{b-}^{\alpha,\psi}f(t) = I_{b-}^{1-\alpha,\psi}\left(-\frac{1}{\psi'(t)}\frac{d}{dt}\right)f(t) = \frac{-1}{\Gamma(1-\alpha)}\int_t^b (\psi(\xi) - \psi(t))^{-\alpha}f'(\xi)d\xi. \tag{7}$$

*Notice that the* $\psi$-Caputo *fractional derivative can reduce to the classical Caputo fractional derivative by choosing the kernel* $\psi(t) = t$.

**Definition 2** ([9]). *If* $f(t) \in C^n[a, b]$, $-\infty \le a < b \le +\infty$, $\alpha > 0$, $n = [\alpha] + 1$ *for* $\alpha \notin \mathbb{N}$, $n = \alpha$ *for* $\alpha \in \mathbb{N}$, *then*

$$^C D_{a+}^{\alpha,\psi}f(t) = D_{a+}^{\alpha,\psi}\left[f(t) - \Sigma_{k=0}^{n-1}\frac{1}{k!}(\psi(t) - \psi(a))^k\left(\frac{1}{\psi'(t)}\frac{d}{dt}\right)^k f(a)\right],$$

$$^C D_{b-}^{\alpha,\psi}f(t) = D_{b-}^{\alpha,\psi}\left[f(t) - \Sigma_{k=0}^{n-1}\frac{(-1)^k}{k!}(\psi(b) - \psi(t))^k\left(\frac{1}{\psi'(t)}\frac{d}{dt}\right)^k f(b)\right].$$

In what follows, we will begin the process of building an appropriate variational structure for the impulsive FDE (1). Before that, a fractional derivative space needs to be established.

**Definition 3.** *Define the* $\psi$-Caputo *fractional derivative space* $E_0^{\alpha,\psi}$ *by the closure of* $C_0^\infty([0, T], \mathbb{R})$ *with weighted norm:*

$$\|x\|_{\alpha,\psi} := \left(\int_0^T |x(t)|^2\, dt + \int_0^T \psi'(t)\, |^C D_{0+}^{\alpha,\psi}x(t)|^2\, dt\right)^{\frac{1}{2}}. \tag{8}$$

*Obviously, the space $E_0^{\alpha,\psi}$ implies that $x(t) \in L^2[0,T]$ with ${}^C D_{0^+}^{\alpha,\psi} x(t) \in L^2[0,T]$, and $x(0) = x(T) = 0$.*

**Lemma 1** ([11])**.** *The space $E_0^{\alpha,\psi}$ is a reflexive and separable Banach space.*

**Lemma 2.** *For any $x(t) \in E_0^{\alpha,\psi}$, $\frac{1}{2} < \alpha \le 1$, we have*

$$\|x\|_\infty \le M \left( \int_0^T \psi'(t) \mid D_{0^+}^{\alpha,\psi} x(t) \mid^2 dt \right)^{\frac{1}{2}}, \tag{9}$$

$$\|x\|_{L^2} \le \widehat{M} \| D_{0^+}^{\alpha,\psi} x \|_{L^2}, \tag{10}$$

*where*

$$M = \frac{(\psi(T) - \psi(0))^{\alpha - \frac{1}{2}}}{\Gamma(\alpha)(2(\alpha - 1) + 1)^{\frac{1}{2}}}, \quad \widehat{M} = \frac{\max_{t \in [0,T]}\{\psi'(t)\}(\psi(T))^\alpha}{\Gamma(\alpha + 1)}.$$

**Proof.** Based on Theorem 4 in [9] and the Hölder inequality, we deduce:

$$|x(t)| = |I_{0^+}^{\alpha,\psi} D_{0^+}^{\alpha,\psi} x(t)| = \frac{1}{\Gamma(\alpha)} \left| \int_0^t \psi'(\xi)(\psi(t) - \psi(\xi))^{\alpha - 1} D_{0^+}^{\alpha,\psi} x(\xi) d\xi \right|$$

$$\le \frac{1}{\Gamma(\alpha)} \left( \int_0^T \left[ (\psi'(\xi))^{\frac{1}{2}}(\psi(t) - \psi(\xi))^{\alpha - 1} \right]^2 d\xi \right)^{\frac{1}{2}} \left( \int_0^T \left[ (\psi'(\xi))^{\frac{1}{2}} D_{0^+}^{\alpha,\psi} x(\xi) \right]^2 d\xi \right)^{\frac{1}{2}}$$

$$\le \frac{(\psi(T) - \psi(0))^{\alpha - \frac{1}{2}}}{\Gamma(\alpha)(2(\alpha - 1) + 1)^{\frac{1}{2}}} \left( \int_0^T \psi'(t)|D_{0^+}^{\alpha,\psi} x(t)|^2 dt \right)^{\frac{1}{2}}.$$

The inequality (10) is immediately available according to [11]. The proof is completed. □

**Lemma 3.** *Based on Definition 2 and $x(0) = x(T) = 0$, one obtains:*

$${}^C D_{0^+}^{\alpha,\psi} x(t) = D_{0^+}^{\alpha,\psi} x(t), \ {}^C D_{T^-}^{\alpha,\psi} x(t) = D_{T^-}^{\alpha,\psi} x(t), \ \forall 0 < \alpha < 1.$$

*From (10) and Lemma 3, we confirm that the norm defined by (8) is equivalent to:*

$$\|x\|_{\alpha,\psi} := \left( \int_0^T \psi'(t) \mid {}^C D_{0^+}^{\alpha,\psi} x(t) \mid^2 dt \right)^{\frac{1}{2}}, \ \forall \, x(t) \in E_0^{\alpha,\psi}. \tag{11}$$

**Lemma 4** ([11])**.** *Let $\frac{1}{2} < \alpha \le 1$. If any sequence $\{x_k\}$ converges to $x$ in $E_0^{\alpha,\psi}$ weakly, then $x_k \to x$ in $C[0,T]$ as $k \to \infty$, i.e., $\|x_k - x\|_\infty \to 0$ as $k \to \infty$.*

*Based on the relevant definitions and lemmas introduced above, the definition of the weak solution of FDE (1) can be given as follows.*

**Lemma 5.** *We say that $x(t) \in E_0^{\alpha,\psi}$ is a weak solution of FDE (1) if the following relationship holds:*

$$\int_0^T \psi'(t) {}^C D_{0^+}^{\alpha,\psi} x(t) {}^C D_{0^+}^{\alpha,\psi} y(t) dt - \sum_{i=1}^n I_i(x(t_i)) y(t_i) = \lambda \sum_{i=0}^n \int_{s_i}^{t_{i+1}} f_i(t, x(t)) \psi'(t) y(t) dt, \forall y(t) \in E_0^{\alpha,\psi}. \tag{12}$$

**Proof.** In view of (6), Dirichlet's formula and Lemma 3 yields:

$$
\int_0^T \psi'(t)^C D_{0+}^{\alpha,\psi} x(t)^C D_{0+}^{\alpha,\psi} y(t) dt = \frac{1}{\Gamma(1-\alpha)} \int_0^T \int_0^t \psi'(t)^C D_{0+}^{\alpha,\psi} x(t)(\psi(t)-\psi(\xi))^{-\alpha} y'(\xi) d\xi dt
$$

$$
= \frac{1}{\Gamma(1-\alpha)} \int_0^T \left[ \int_t^T \psi'(\xi)^C D_{0+}^{\alpha,\psi} x(\xi)(\psi(\xi)-\psi(t))^{-\alpha} d\xi \right] y'(t) dt
$$

$$
= \frac{1}{\Gamma(1-\alpha)} \sum_{i=0}^n \int_{s_i}^{t_{i+1}} \left[ \int_t^T \psi'(\xi) D_{0+}^{\alpha,\psi} x(\xi)(\psi(\xi)-\psi(t))^{-\alpha} d\xi \right] y'(t) dt
$$

$$
+ \frac{1}{\Gamma(1-\alpha)} \sum_{i=1}^n \int_{t_i}^{s_i} \left[ \int_t^T \psi'(\xi) D_{0+}^{\alpha,\psi} x(\xi)(\psi(\xi)-\psi(t))^{-\alpha} d\xi \right] y'(t) dt. \tag{13}
$$

Due to (4), (5) and (7) yields

$$
\frac{1}{\Gamma(1-\alpha)} \sum_{i=0}^n \int_{s_i}^{t_{i+1}} \left[ \int_t^T \psi'(\xi) D_{0+}^{\alpha,\psi} x(\xi)(\psi(\xi)-\psi(t))^{-\alpha} d\xi \right] y'(t) dt \tag{14}
$$

$$
= \frac{1}{\Gamma(1-\alpha)} \sum_{i=0}^n \left[ \int_t^T \psi'(\xi) D_{0+}^{\alpha,\psi} x(\xi)(\psi(\xi)-\psi(t))^{-\alpha} d\xi \right] y(t) \Big|_{t=s_i^+}^{t=t_{i+1}^-}
$$

$$
- \frac{1}{\Gamma(1-\alpha)} \sum_{i=0}^n \int_{s_i}^{t_{i+1}} \frac{d}{dt} \left[ \int_t^T \psi'(\xi) D_{0+}^{\alpha,\psi} x(\xi)(\psi(\xi)-\psi(t))^{-\alpha} d\xi \right] \cdot y(t) dt
$$

$$
= \sum_{i=0}^n \frac{1}{\Gamma(1-\alpha)} \int_t^T \psi'(\xi)(\psi(\xi)-\psi(t))^{-\alpha} \left( \frac{1}{\psi'(\xi)} \frac{d}{d\xi} \right) I_{0+}^{1-\alpha,\psi} x(\xi) d\xi \cdot y(t) \Big|_{t=s_i^+}^{t=t_{i+1}^-}
$$

$$
+ \sum_{i=0}^n \int_{s_i}^{t_{i+1}} \frac{-1}{\Gamma(1-\alpha)} \left( \frac{1}{\psi'(t)} \frac{d}{dt} \right) \left[ \int_t^T \psi'(\xi)(\psi(\xi)-\psi(t))^{-\alpha} D_{0+}^{\alpha,\psi} x(\xi) d\xi \right] \cdot \psi'(t) y(t) dt
$$

$$
= \sum_{i=0}^n -^C D_{T^-}^{\alpha,\psi} (I_{0+}^{1-\alpha,\psi} x(t)) y(t) \Big|_{t=s_i^+}^{t=t_{i+1}^-} + \sum_{i=0}^n \int_{s_i}^{t_{i+1}} D_{T^-}^{\alpha,\psi} (D_{0+}^{\alpha,\psi} x(t)) \psi'(t) y(t) dt,
$$

and

$$
\frac{1}{\Gamma(1-\alpha)} \sum_{i=1}^n \int_{t_i}^{s_i} \left[ \int_t^T \psi'(\xi) D_{0+}^{\alpha,\psi} x(\xi)(\psi(\xi)-\psi(t))^{-\alpha} d\xi \right] y'(t) dt \tag{15}
$$

$$
= \frac{1}{\Gamma(1-\alpha)} \sum_{i=1}^n \left[ \int_t^T \psi'(\xi) D_{0+}^{\alpha,\psi} x(\xi)(\psi(\xi)-\psi(t))^{-\alpha} d\xi \right] y(t) \Big|_{t=t_i^+}^{t=s_i^-}
$$

$$
- \frac{1}{\Gamma(1-\alpha)} \sum_{i=1}^n \int_{t_i}^{s_i} \frac{d}{dt} \left[ \int_t^T \psi'(\xi) D_{0+}^{\alpha,\psi} x(\xi)(\psi(\xi)-\psi(t))^{-\alpha} d\xi \right] \cdot y(t) dt
$$

$$
= \sum_{i=1}^n \frac{1}{\Gamma(1-\alpha)} \int_t^T \psi'(\xi)(\psi(\xi)-\psi(t))^{-\alpha} \left( \frac{1}{\psi'(\xi)} \frac{d}{d\xi} \right) I_{0+}^{1-\alpha,\psi} x(\xi) d\xi \cdot y(t) \Big|_{t=t_i^+}^{t=s_i^-}
$$

$$
+ \sum_{i=1}^n \int_{t_i}^{s_i} \frac{d}{dt} \left[ \frac{-1}{\Gamma(1-\alpha)} \int_t^T \psi'(\xi)(\psi(\xi)-\psi(t))^{-\alpha} \left( \frac{1}{\psi'(\xi)} \frac{d}{d\xi} \right) I_{0+}^{1-\alpha,\psi} x(\xi) d\xi \right] \cdot y(t) dt
$$

$$
= \sum_{i=1}^n -^C D_{T^-}^{\alpha,\psi} (I_{0+}^{1-\alpha,\psi} x(t)) y(t) \Big|_{t=t_i^+}^{t=s_i^-} + \sum_{i=1}^n \int_{t_i}^{s_i} \frac{d}{dt} \left[ ^C D_{T^-}^{\alpha,\psi} (I_{0+}^{1-\alpha,\psi} x(t)) \right] \cdot y(t) dt.
$$

Consequently, combining (13), (14), (15), and the impulsive conditions in FDE (1), one has:

$$\int_0^T \psi'(t)^C D_{0+}^{\alpha,\psi} x(t)^C D_{0+}^{\alpha,\psi} y(t)dt$$

$$= \sum_{i=0}^n -^C D_{T-}^{\alpha,\psi}(I_{0+}^{1-\alpha,\psi} x(t))y(t)\mid_{t=s_i^+}^{t=t_{i+1}^-} + \sum_{i=1}^n -^C D_{T-}^{\alpha,\psi}(I_{0+}^{1-\alpha,\psi} x(t))y(t)\mid_{t=t_i^+}^{t=s_i^-} + \sum_{i=0}^n \int_{s_i}^{t_{i+1}} D_{T-}^{\alpha,\psi}(D_{0+}^{\alpha,\psi} x(t))\psi'(t)y(t)dt$$

$$= \sum_{i=1}^n {}^C D_{T-}^{\alpha,\psi}(I_{0+}^{1-\alpha,\psi} x(t_i^+))y(t_i^+) - {}^C D_{T-}^{\alpha,\psi}(I_{0+}^{1-\alpha,\psi} x(t_i^-))y(t_i^-) + \sum_{i=1}^n {}^C D_{T-}^{\alpha,\psi}(I_{0+}^{1-\alpha,\psi} x(s_i^+))y(s_i^+) - {}^C D_{T-}^{\alpha,\psi}(I_{0+}^{1-\alpha,\psi} x(s_i^-))y(s_i^-) \qquad (16)$$

$$+ {}^C D_{T-}^{\alpha,\psi}(I_{0+}^{1-\alpha,\psi} x(0))y(0) - {}^C D_{T-}^{\alpha,\psi}(I_{0+}^{1-\alpha,\psi} x(T))y(T) + \sum_{i=0}^n \int_{s_i}^{t_{i+1}} D_{T-}^{\alpha,\psi}(D_{0+}^{\alpha,\psi} x(t))\psi'(t)y(t)dt$$

$$= \sum_{i=1}^n I_i(x(t_i))y(t_i) + \sum_{i=0}^n \int_{s_i}^{t_{i+1}} D_{T-}^{\alpha,\psi}(D_{0+}^{\alpha,\psi} x(t))\psi'(t)y(t)dt.$$

An equivalent form for FDE (1) can be derived by multiplying the first equation of (1) with $\psi'(t)y(t)$, and integrating on both sides from $s_i$ to $t_{i+1}$, then summing from $i = 0$ to $i = n$, according to (16), one has:

$$\int_0^T \psi'(t)^C D_{0+}^{\alpha,\psi} x(t)^C D_{0+}^{\alpha,\psi} y(t)dt - \sum_{i=1}^n I_i(x(t_i))y(t_i) = \lambda \sum_{i=0}^n \int_{s_i}^{t_{i+1}} f_i(t,x(t))\psi'(t)y(t)dt.$$

The proof is completed. $\square$

**Definition 4.** *A function*

$$x \in \left\{ x \in AC[0,T]: \int_{s_i}^{t_{i+1}} \mid x(t) \mid^2 + \psi'(t) \mid {}^C D_{0+}^{\alpha,\psi} x(t) \mid^2 dt < +\infty, i = 1, 2, \ldots, n \right\}$$

*is called a classical solution of FDE (1) if $x$ satisfies the first equation of FDE (1), the limits ${}^C D_{T-}^{\alpha,\psi}(I_{0+}^{1-\alpha,\psi} x)(t_i^{\pm})$ and ${}^C D_{T-}^{\alpha,\psi}(I_{0+}^{1-\alpha,\psi} x)(s_i^{\pm})$ exist and satisfy the impulsive conditions in (1), and boundary condition $x(0) = x(T) = 0$ holds.*

**Lemma 6** ([23]). *Let $E$ be a real reflexive Banach space, let $J_1 : E \to \mathbb{R}$ be a sequentially weakly lower semi-continuous, coercive and continuously Gâteaux differentiable functional whose Gâteaux derivative admits a continuous inverse on $E^*$, and let $J_2 : E \to \mathbb{R}$ be a sequentially weakly upper semi-continuous and continuously Gâteaux differentiable functional whose Gâteaux derivative is compact. Suppose that there exist $\rho \in \mathbb{R}$ and $x_1 \in E$ with $0 < \rho < J_1(x_1)$, such that*
*(i) $\sup_{x \in J_1^{-1}(]-\infty,\rho])} J_2(x) < \rho \frac{J_2(x_1)}{J_1(x_1)}$.*

*(ii) For all $\lambda \in \mathcal{B} := \left] \frac{J_1(x_1)}{J_2(x_1)}, \frac{\rho}{\sup_{x \in J_1^{-1}(]-\infty,\rho])} J_2(x)} \right[$, the functional $J_1 - \lambda J_2$ is coercive.*

*Then, for each $\lambda \in \mathcal{B}$, the functional $J_1 - \lambda J_2$ possesses at least three distinct critical points on $E$.*

## 3. Proof of Theorems

In this section, the multiplicity of at least three distinct classical solutions for impulsive FDE (1) is discussed depending on Lemma 6 and Definition 4.

For any $x(t) \in E_0^{\alpha,\psi}$, define the functional $J_\lambda := J_1 - \lambda J_2$, where

$$J_1(x) = \frac{1}{2} \int_0^T \psi'(t)|^C D_{0+}^{\alpha,\psi} x(t)|^2 dt - \sum_{i=1}^n \int_0^{x(t_i)} I_i(\xi)d\xi,$$

$$J_2(x) = \sum_{i=0}^n \int_{s_i}^{t_{i+1}} F_i(t,x(t))\psi'(t)dt, \qquad (17)$$

where $F_i(t,x) = \int_0^x f_i(t,\xi)d\xi$. Owing to the continuity of $f_i$ and $I_i$, we can obtain $J_1, J_2 \in C^1(E_0^{\alpha,\psi}, \mathbb{R})$ and

$$J_1'(x)(y) = \int_0^T \psi'(t)^C D_{0+}^{\alpha,\psi} x(t)^C D_{0+}^{\alpha,\psi} y(t)dt - \sum_{i=1}^n I_i(x(t_i))y(t_i),$$

$$J_2'(x)(y) = \sum_{i=0}^n \int_{s_i}^{t_{i+1}} f_i(t,x(t))\psi'(t)y(t)dt. \tag{18}$$

Apparently, the critical point of $J_\lambda$ is the weak solution of impulsive FDE (1).

**Theorem 1.** *Assume that*
$(A_1)$ $I_i(0) = 0$ *and there exist* $d_i, L_i > 0$ *with* $\max\{M^2 \sum_{i=1}^n L_i, M^2 \sum_{i=1}^n d_i\} < 1$, *such that* $|I_i(\xi)| \le d_i|\xi|$ *and* $| I_i(\xi_1) - I_i(\xi_2) | \le L_i | \xi_1 - \xi_2 |, \forall \xi, \xi_1, \xi_2 \in \mathbb{R}$.
$(A_2)$ *There exist a constant* $\rho > 0$ *and a function* $\varsigma(t)$, *such that* $\left(\frac{1}{2} - \frac{M^2 \sum_{i=1}^n d_i}{2}\right) \|\varsigma\|_{\alpha,\psi}^2 > \rho$,
*and*

$$\frac{\sum_{i=0}^n \int_{s_i}^{t_{i+1}} \sup_{x \in \Omega_\rho} F_i(t,x(t))\psi'(t)dt}{\rho} < \frac{2 \sum_{i=0}^n \int_{s_i}^{t_{i+1}} F_i(t,\varsigma(t))\psi'(t)dt}{\|\varsigma\|_{\alpha,\psi}^2 - 2\sum_{i=1}^n \int_0^{\varsigma(t_i)} I_i(\xi)d\xi}, \tag{19}$$

*where* $\Omega_\rho = \{x \in \mathbb{R} : \left(\frac{1}{2M^2} - \frac{\sum_{i=1}^n d_i}{2}\right) | x |^2 \le \rho\}$.
$(A_3)$ *there exist* $b_i, c_i > 0$, $\theta_i \in [0,1)$, *such that* $|f_i(t,x)| \le b_i + c_i|x|^{\theta_i}, \forall t \in [0,T], x \in \mathbb{R}$, $i = 0, 1, \ldots, n$.
*Then, for each* $\lambda \in \left] \frac{\|\varsigma\|_{\alpha,\psi}^2 - 2\sum_{i=1}^n \int_0^{\varsigma(t_i)} I_i(\xi)d\xi}{2\sum_{i=0}^n \int_{s_i}^{t_{i+1}} F_i(t,\varsigma(t))\psi'(t)dt}, \frac{\rho}{\sum_{i=0}^n \int_{s_i}^{t_{i+1}} \sup_{x \in \Omega_\rho} F_i(t,x(t))\psi'(t)dt} \right[$, *the impulsive*
FDE (1) *possesses at least three distinct weak solutions on* $E_0^{\alpha,\psi}$.

**Proof.** First, we are concerned with functionals $J_1$ and $J_2$. Let $\{x_k\}_{k=1}^\infty$ be a weakly convergent sequence to $x$ in $E_0^{\alpha,\psi}$, then $\|x\|_{\alpha,\psi} \le \liminf_{k\to\infty} \|x_k\|_{\alpha,\psi}$. In view of Lemma 4 that $\{x_k\}$ converges to $x$ in $C([0,T], \mathbb{R})$ uniformly. That is:

$$\liminf_{k\to\infty} J_1(x_k) = \liminf_{k\to\infty} \left\{ \frac{1}{2}\|x_k\|_{\alpha,\psi}^2 - \sum_{i=1}^n \int_0^{x_k(t_i)} I_i(\xi)d\xi \right\}$$

$$\ge \frac{1}{2}\|x\|_{\alpha,\psi}^2 - \sum_{i=1}^n \int_0^{x(t_i)} I_i(\xi)d\xi = J_1(x),$$

which means that $J_1$ is weakly lower semi-continuous. In what follows, we assert that $J_1$ possesses a continuous inverse on $(E_0^{\alpha,\psi})^*$. By means of (18), (9) and $(A_1)$ yield:

$$(J_1'(x) - J_1'(y))(x - y) = \int_0^T \psi'(t) | {}^C D_{0+}^{\alpha,\psi}(x(t) - y(t)) |^2 dt - \sum_{i=1}^n (I_i(x(t_i)) - I_i(y(t_i)))(x(t_i) - y(t_i))$$

$$\ge \|x - y\|_{\alpha,\psi}^2 - \sum_{i=1}^n | I_i(x(t_i)) - I_i(y(t_i)) || x(t_i) - y(t_i) |$$

$$\ge \|x - y\|_{\alpha,\psi}^2 - \sum_{i=1}^n L_i | x(t_i) - y(t_i) |^2$$

$$\ge \|x - y\|_{\alpha,\psi}^2 - \|x - y\|_\infty^2 \sum_{i=1}^n L_i$$

$$\ge (1 - M^2 \sum_{i=1}^n L_i)\|x - y\|_{\alpha,\psi}^2 > 0, \forall x \ne y,$$

which shows that $J_1'$ is strictly monotone. Based on the Theorem 26.A(d) in [24], we can obtain that there exists an inverse of $J_1'$ on $(E_0^{\alpha,\psi})^*$, and the inverse is continuous. Obviously, $J_1$ is coercive. On the other hand, suppose that $\{x_k\} \subset E_0^{\alpha,\psi}$, $x_k \rightharpoonup x$ in $E_0^{\alpha,\psi}$ as $k \to \infty$. Then, $x_k \to x$ uniformly on $[0, T]$, and

$$\limsup_{k\to\infty} J_2(x_k) \le \sum_{i=0}^{n} \int_{s_i}^{t_{i+1}} \limsup_{k\to\infty} F_i(t, x_k(t))\psi'(t)dt = \sum_{i=0}^{n} \int_{s_i}^{t_{i+1}} F_i(t, x(t))\psi'(t)dt = J_2(x),$$

hence, $J_2$ is sequentially weakly upper semi-continuous. Considering $F_i \in C^1((s_i, t_{i+1}] \times \mathbb{R}, \mathbb{R})$, then $F_i(t, x_k(t)) \to F_i(t, x(t))$ as $k \to \infty$. According to the Lebesgue control convergence theorem, $J_2'(x_k) \to J_2'(x)$, i.e., $J_2'$ is continuous strongly on $E_0^{\alpha,\psi}$. So, $J_2'$ is a compact operator.

Take $x_0 = 0$, $x_1 = \varsigma$. Due to $(A_1)$ and $(A_2)$, we have $J_1(x_1) \ge \left(\frac{1}{2} - \frac{M^2\sum_{i=1}^{n} d_i}{2}\right)\|x_1\|_{\alpha,\psi}^2 > \rho > 0$ and $J_1(x_0) = 0$. In view of (17), (9), and $(A_1)$, we have:

$$J_1^{-1}(]-\infty, \rho]) = \{x \in E_0^{\alpha,\psi} : J_1(x) \le \rho\} = \{x \in E_0^{\alpha,\psi} : \frac{1}{2}\int_0^T \psi'(t)|^C D_{0+}^{\alpha,\psi}x(t)|^2 dt - \sum_{i=1}^{n} \int_0^{x(t_i)} I_i(\xi))d\xi \le \rho\}$$

$$\subseteq \{x \in E_0^{\alpha,\psi} : \frac{1}{2}\|x\|_{\alpha,\psi}^2 - \sum_{i=1}^{n} \int_0^{x(t_i)} d_i|\xi|d\xi \le \rho\}$$

$$\subseteq \{x \in E_0^{\alpha,\psi} : \left(\frac{1}{2M^2} - \frac{\sum_{i=1}^{n} d_i}{2}\right) \mid x(t) \mid^2 \le \rho, t \in [0, T]\},$$

then

$$\sup_{x \in J_1^{-1}(]-\infty,\rho])} J_2(x) = \sup_{x \in J_1^{-1}(]-\infty,\rho])} \sum_{i=0}^{n} \int_{s_i}^{t_{i+1}} F_i(t, x(t))\psi'(t)dt \le \sum_{i=0}^{n} \int_{s_i}^{t_{i+1}} \sup_{x \in \Omega_\rho} F_i(t, x(t))\psi'(t)dt,$$

that is

$$\frac{\sup_{x \in J_1^{-1}(]-\infty,\rho])} J_2(x)}{\rho} \le \frac{\sum_{i=0}^{n} \int_{s_i}^{t_{i+1}} \sup_{x \in \Omega_\rho} F_i(t, x(t))\psi'(t)dt}{\rho} < \frac{2\sum_{i=0}^{n} \int_{s_i}^{t_{i+1}} F_i(t, \varsigma(t))\psi'(t)dt}{\|\varsigma\|_{\alpha,\psi}^2 - 2\sum_{i=1}^{n} \int_0^{\varsigma(t_i)} I_i(\xi)d\xi} = \frac{J_2(x_1)}{J_1(x_1)},$$

where (27) is used. Thus, the assumption (i) of Lemma 6 is satisfied.

In addition, for any fixed $\lambda \in \mathcal{B}$, by means of (17), $(A_1)$, $(A_3)$, and (9), we obtain:

$$J_1(x) - \lambda J_2(x) \ge \frac{1}{2}\|x\|_{\alpha,\psi}^2 - \sum_{i=1}^{n} \left(\int_0^{x(t_i)} d_i|\xi|d\xi\right) - \lambda \sum_{i=0}^{n} \int_{s_i}^{t_{i+1}} \psi'(t) \int_0^x b_i + c_i|s|^{\theta_i} ds dt$$

$$\ge \frac{1}{2}\|x\|_{\alpha,\psi}^2 - \left(\frac{1}{2}\|x\|_\infty^2 \sum_{i=1}^{n} d_i\right) - \lambda(\psi(T) - \psi(0))\left(\sum_{i=0}^{n} b_i\|x\|_\infty + \frac{c_i}{\theta_i + 1}\|x\|_\infty^{\theta_i+1}\right)$$

$$\ge \left(\frac{1}{2} - \frac{M^2\sum_{i=1}^{n} d_i}{2}\right)\|x\|_{\alpha,\psi}^2 - \lambda(\psi(T) - \psi(0))M\|x\|_{\alpha,\psi}\left(\sum_{i=0}^{n} b_i\right)$$

$$- \lambda(\psi(T) - \psi(0)) \sum_{i=0}^{n} \frac{c_i M^{\theta_i+1}}{\theta_i + 1}\|x\|_{\alpha,\psi}^{\theta_i+1}.$$

Since $\theta_i \in [0, 1)$ and $M^2\sum_{i=1}^{n} d_i < 1$, we assert that $\lim_{\|x\|_{\alpha,\psi}\to\infty} J_1(x) - \lambda J_2(x) = +\infty$, which implies that $J_1 - \lambda J_2$ is coercive. The condition (ii) in Lemma 6 holds. Consequently, the impulsive FDE (1) possesses at least three distinct weak solutions on $E_0^{\alpha,\psi}$ using Lemma 6. □

**Theorem 2.** *$x(t)$ is a weak solution of impulsive FDE (1), if and only if $x(t)$ is a classical solution of FDE (1).*

**Proof.** If $x(t)$ is a classical solution of impulsive FDE (1), then $x(t)$ also is a weak solution obviously. On the other hand, if $x(t) \in E_0^{\alpha,\psi}$ is a weak solution of FDE (1), then $x(0) = x(T) = 0$ and the Equation (12) holds. Without loss of generality, choose a test function $v_i(t) \in C_0^\infty(s_i, t_{i+1}]$ and $v_i(t) \equiv 0$ for $t \in [0, s_i] \bigcup (t_{i+1}, T]$, $i = 0, 2, \ldots, n$. Substituting $v_i(t)$ into (12), from (16), we have:

$$\int_{s_i}^{t_{i+1}} D_{T^-}^{\alpha,\psi}(D_{0^+}^{\alpha,\psi}x(t))\psi'(t)v_i(t)dt = \int_{s_i}^{t_{i+1}} \psi'(t)\,{}^C D_{0^+}^{\alpha,\psi}x(t)\,{}^C D_{0^+}^{\alpha,\psi}v_i(t)dt,$$

$$\int_{s_i}^{t_{i+1}} \psi'(t)\,{}^C D_{0^+}^{\alpha,\psi}x(t)\,{}^C D_{0^+}^{\alpha,\psi}v_i(t)dt = \lambda \int_{s_i}^{t_{i+1}} f_i(t, x(t))\psi'(t)v_i(t)dt,$$

which shows that

$$^C D_{T^-}^{\alpha,\psi}(^C D_{0^+}^{\alpha,\psi}x(t)) = \lambda f_i(t, x(t)), \forall t \in [s_i, t_{i+1}], i = 0, 1, \ldots, n. \tag{20}$$

Because $x \in E_0^{\alpha,\psi} \subset C[0, T]$ and $\psi(t) \in C^1[0, T]$, then

$$\int_{s_i}^{t_{i+1}} \mid x(t) \mid^2 + \psi'(t) \mid {}^C D_{0^+}^{\alpha,\psi}x(t) \mid^2 dt < +\infty.$$

Based on Lemma 3, (4) and (7) yield:

$$
\begin{aligned}
^C D_{T^-}^{\alpha,\psi}(^C D_{0^+}^{\alpha,\psi}x(t)) &= D_{T^-}^{\alpha,\psi}(D_{0^+}^{\alpha,\psi}x(t)) = D_{T^-}^{\alpha,\psi}\left[\frac{1}{\psi'(t)}\frac{d}{dt}I_{0^+}^{1-\alpha,\psi}x(t)\right]\\
&= \frac{-1}{\Gamma(1-\alpha)}\left(\frac{1}{\psi'(t)}\frac{d}{dt}\right)\int_t^T \psi'(\xi)(\psi(\xi)-\psi(t))^{-\alpha}\left(\frac{1}{\psi'(\xi)}\frac{d}{d\xi}\right)I_{0^+}^{1-\alpha,\psi}x(\xi)d\xi\\
&= \frac{1}{\psi'(t)}\frac{d}{dt}\left[^C D_{T^-}^{\alpha,\psi}I_{0^+}^{1-\alpha,\psi}x(t)\right].
\end{aligned}
\tag{21}
$$

Since $\psi(t) \in C^1[0, T]$, $f_i \in C((s_i, t_{i+1}] \times \mathbb{R}, \mathbb{R})$, according to (20) and (21), one obtains $^C D_{T^-}^{\alpha,\psi}I_{0^+}^{1-\alpha,\psi}x(t) \in AC[s_i, t_{i+1}]$, which implies that the following limits exist:

$$^C D_{T^-}^{\alpha,\psi}(I_{0^+}^{1-\alpha,\psi}x)(s_i^+) = \lim_{t \to s_i^+} {}^C D_{T^-}^{\alpha,\psi}(I_{0^+}^{1-\alpha,\psi}x)(t),$$

$$^C D_{T^-}^{\alpha,\psi}(I_{0^+}^{1-\alpha,\psi}x)(t_{i+1}^-) = \lim_{t \to t_{i+1}^-} {}^C D_{T^-}^{\alpha,\psi}(I_{0^+}^{1-\alpha,\psi}x)(t).$$

Substituting (20) into (12), one obtains:

$$\int_0^T \psi'(t)\,{}^C D_{0^+}^{\alpha,\psi}x(t)\,{}^C D_{0^+}^{\alpha,\psi}y(t)dt - \sum_{i=1}^n I_i(x(t_i))y(t_i) - \sum_{i=0}^n \int_{s_i}^{t_{i+1}} {}^C D_{T^-}^{\alpha,\psi}(^C D_{0^+}^{\alpha,\psi}x(t))\psi'(t)y(t)dt = 0. \tag{22}$$

Uniting (13) with (14), we have:

$$\int_0^T \psi'(t)\,^C D_{0+}^{\alpha,\psi} x(t)\,^C D_{0+}^{\alpha,\psi} y(t)\,dt$$

$$= \sum_{i=0}^n \int_{s_i}^{t_{i+1}} \psi'(t)\,^C D_{0+}^{\alpha,\psi} x(t)\,^C D_{0+}^{\alpha,\psi} y(t)\,dt + \sum_{i=1}^n \int_{t_i}^{s_i} \psi'(t)\,^C D_{0+}^{\alpha,\psi} x(t)\,^C D_{0+}^{\alpha,\psi} y(t)\,dt$$

$$= \sum_{i=0}^n {}^C D_{T-}^{\alpha,\psi}(I_{0+}^{1-\alpha,\psi} x(s_i^+))y(s_i^+) - \sum_{i=0}^n {}^C D_{T-}^{\alpha,\psi}(I_{0+}^{1-\alpha,\psi} x(t_{i+1}^-))y(t_{i+1}^-) \qquad (23)$$

$$+ \sum_{i=0}^n \int_{s_i}^{t_{i+1}} D_{T-}^{\alpha,\psi}(D_{0+}^{\alpha,\psi} x(t))\psi'(t)y(t)\,dt + \sum_{i=1}^n \int_{t_i}^{s_i} \psi'(t)\,^C D_{0+}^{\alpha,\psi} x(t)\,^C D_{0+}^{\alpha,\psi} y(t)\,dt.$$

Then, from (22) and (23), we obtain:

$$\sum_{i=0}^n {}^C D_{T-}^{\alpha,\psi}(I_{0+}^{1-\alpha,\psi} x(s_i^+))y(s_i^+) - \sum_{i=0}^n {}^C D_{T-}^{\alpha,\psi}(I_{0+}^{1-\alpha,\psi} x(t_{i+1}^-))y(t_{i+1}^-)$$

$$+ \sum_{i=1}^n \int_{t_i}^{s_i} \psi'(t)\,^C D_{0+}^{\alpha,\psi} x(t)\,^C D_{0+}^{\alpha,\psi} y(t)\,dt - \sum_{i=1}^n I_i(x(t_i))y(t_i) = 0. \qquad (24)$$

Without loss of generality, assume $v_i(t) \in C_0^\infty(t_i, s_i]$ and $v_i(t) \equiv 0$ for $t \in [0, t_i] \bigcup (s_i, T]$, $i = 1, 2, \ldots, n$. Substituting $v_i(t)$ into (24), from (15) we deduce:

$$\sum_{i=1}^n \int_{t_i}^{s_i} \frac{d}{dt}\left[{}^C D_{T-}^{\alpha,\psi}(I_{0+}^{1-\alpha,\psi} x(t))\right] v_i(t)\,dt = 0,$$

because of the arbitrariness of $v_i(t)$, for $t \in (t_i, s_i]$, $i = 1, 2, \ldots, n$, we can obtain ${}^C D_{T-}^{\alpha,\psi}(I_{0+}^{1-\alpha,\psi} x(t)) = Constant$. That is:

$$^C D_{T-}^{\alpha,\psi}(I_{0+}^{1-\alpha,\psi} x)(t) = {}^C D_{T-}^{\alpha,\psi}(I_{0+}^{1-\alpha,\psi} x)(t_i^+) = {}^C D_{T-}^{\alpha,\psi}(I_{0+}^{1-\alpha,\psi} x)(s_i^-), \ t \in (t_i, s_i], \ i = 1, 2, \ldots, n. \qquad (25)$$

Substituting (25) back into (24) yields:

$$\sum_{i=0}^n {}^C D_{T-}^{\alpha,\psi}(I_{0+}^{1-\alpha,\psi} x(s_i^+))y(s_i^+) - \sum_{i=0}^n {}^C D_{T-}^{\alpha,\psi}(I_{0+}^{1-\alpha,\psi} x(t_{i+1}^-))y(t_{i+1}^-) - \sum_{i=1}^n I_i(x(t_i))y(t_i)$$

$$+ \sum_{i=1}^n {}^C D_{T-}^{\alpha,\psi}(I_{0+}^{1-\alpha,\psi} x(t_i^+))y(t_i) - \sum_{i=1}^n {}^C D_{T-}^{\alpha,\psi}(I_{0+}^{1-\alpha,\psi} x(t_i^+))y(s_i) = 0,$$

then

$$\sum_{i=1}^n \left[{}^C D_{T-}^{\alpha,\psi}(I_{0+}^{1-\alpha,\psi} x(t_i^+)) - {}^C D_{T-}^{\alpha,\psi}(I_{0+}^{1-\alpha,\psi} x(t_i^-)) - I_i(x(t_i))\right] y(t_i)$$

$$+ \sum_{i=1}^n \left[{}^C D_{T-}^{\alpha,\psi}(I_{0+}^{1-\alpha,\psi} x(s_i^+)) - {}^C D_{T-}^{\alpha,\psi}(I_{0+}^{1-\alpha,\psi} x(t_i^+))\right] y(s_i) = 0,$$

which implies that

$$^C D_{T-}^{\alpha,\psi}(I_{0+}^{1-\alpha,\psi} x(t_i^+)) - {}^C D_{T-}^{\alpha,\psi}(I_{0+}^{1-\alpha,\psi} x(t_i^-)) = I_i(x(t_i)), \ ^C D_{T-}^{\alpha,\psi}(I_{0+}^{1-\alpha,\psi} x(s_i^+)) = {}^C D_{T-}^{\alpha,\psi}(I_{0+}^{1-\alpha,\psi} x(t_i^+)).$$

Combining with (25), we can obtain ${}^C D_{T-}^{\alpha,\psi}(I_{0+}^{1-\alpha,\psi} x(s_i^+)) = {}^C D_{T-}^{\alpha,\psi}(I_{0+}^{1-\alpha,\psi} x(s_i^-))$ for $i = 1, 2, \ldots, n$. Consequently, boundary conditions and impulsive conditions, as well as the first equation in FDE (1), are all satisfied by $x(t)$, which shows that $x(t)$ is a classical solution of FDE (1). $\square$

**Example 1.** *Let $\alpha = 0.6$, $\psi(t) = e^t$, $t \in [0,1]$. Concern with the following system is as follows:*

$$
\begin{cases}
{}^C D_{1^-}^{0.6,e^t}({}^C D_{0^+}^{0.6,e^t} x(t)) = \lambda x^{\frac{1}{5}}(t), \ t \in (0,t_1] \bigcup (s_1,1], \\
\Delta({}^C D_{1^-}^{0.6,e^t}(I_{0^+}^{0.4,e^t} x))(t_1) = I_1(x(t_1)), \\
{}^C D_{1^-}^{0.6,e^t}(I_{0^+}^{0.4,e^t} x)(t) = {}^C D_{1^-}^{0.6,e^t}(I_{0^+}^{0.4,e^t} x)(t_1^+), \ \ t \in (t_1,s_1], \\
{}^C D_{1^-}^{0.6,e^t}(I_{0^+}^{0.4,e^t} x)(s_1^-) = {}^C D_{1^-}^{0.6,e^t}(I_{0^+}^{0.4,e^t} x)(s_1^+), \\
x(0) = x(1) = 0.
\end{cases}
\tag{26}
$$

*Put $I_1(x) = \frac{1}{100}x$. Clearly, $d_1 = L_1 = \frac{1}{100}$. By direct calculation, we have $M \approx 1.585$, $M^2 L_1 = M^2 d_1 \approx 0.025$, the condition $(A_1)$ in Theorem 1 holds. Choose $\varsigma(t) = \Gamma(1.2)e^t$, $\rho = \frac{1}{10}$, a direct calculation yields*

$$
{}^C D_{0^+}^{0.6,e^t} \varsigma(t) = \frac{\Gamma(1.2)}{\Gamma(0.4)}(-\frac{5}{2})(e^t - 1)^{0.4}, \ \|\varsigma\|_{\alpha,\psi}^2 \approx 1.576, \ \left(\frac{1}{2} - \frac{M^2 d_1}{2}\right)\|\varsigma\|_{\alpha,\psi}^2 \approx 0.8 > \rho,
$$

*then*

$$
\frac{\sum_{i=0}^n \int_{s_i}^{t_{i+1}} \sup_{x \in \Omega(\rho)} F_i(t,x(t))\psi'(t)dt}{\rho} = \frac{(\int_0^{t_1} + \int_{s_1}^1)\frac{5}{6}e^t \sup_{x \in \Omega(\rho)} x^{\frac{6}{5}}(t)dt}{0.1} \approx 0.55\left(\int_0^{t_1} + \int_{s_1}^1\right)e^t dt < 0.9,
$$

*and*

$$
\frac{2\sum_{i=0}^n \int_{s_i}^{t_{i+1}} F_i(t,\varsigma(t))\psi'(t)dt}{\|\varsigma\|_{\alpha,\psi}^2 - 2\sum_{i=1}^n \int_0^{\varsigma(t_i)} I_i(s))ds} = \frac{\frac{5}{3}(\int_0^{t_1} + \int_{s_1}^1)e^t(\Gamma(1.2)e^t)^{\frac{6}{5}}dt}{\|\varsigma\|_{\alpha,\psi}^2 - \frac{1}{100}(\varsigma(t_1))^2}
$$

$$
> \frac{\frac{5}{3}(\int_0^{t_1} + \int_{s_1}^1)e^t(\Gamma(1.2)e^t)^{\frac{6}{5}}dt}{\|\varsigma\|_{\alpha,\psi}^2 - \frac{1}{100}(\Gamma(1.2))^2} \approx 1.2\left(\int_0^{t_1} + \int_{s_1}^1\right)(e^t)^{\frac{11}{5}}dt > 1.2,
$$

*which shows that the condition $(A_2)$ holds. From Theorem 1, the system (26) possesses at least three distinct classical solutions for each $\lambda \in ]0.8, 1.1[$.*

**Example 2.** *Let $\alpha = 0.75$, $\psi(t) = ct^\sigma$ with $\sigma > 0$ and $c \geq 1$, $t \in [0,1]$. Concern with the following system is as follows:*

$$
\begin{cases}
{}^C D_{1^-}^{0.75,ct^\sigma}({}^C D_{0^+}^{0.75,ct^\sigma} x(t)) = \lambda f(t,x(t)), \ t \in (0,t_1] \bigcup (s_1,1], \\
\Delta({}^C D_{1^-}^{0.75,ct^\sigma}(I_{0^+}^{0.25,ct^\sigma} x))(t_1) = I_1(x(t_1)), \\
{}^C D_{1^-}^{0.75,ct^\sigma}(I_{0^+}^{0.25,ct^\sigma} x)(t) = {}^C D_{1^-}^{0.75,ct^\sigma}(I_{0^+}^{0.25,ct^\sigma} x)(t_1^+), \ \ t \in (t_1,s_1], \\
{}^C D_{1^-}^{0.75,ct^\sigma}(I_{0^+}^{0.25,ct^\sigma} x)(s_1^-) = {}^C D_{1^-}^{0.75,ct^\sigma}(I_{0^+}^{0.25,ct^\sigma} x)(s_1^+), \\
x(0) = x(1) = 0.
\end{cases}
\tag{27}
$$

*Obviously, if one chooses $c = 1$, i.e., $\psi(t) = t^\sigma$, the system (27) can reduce into the well known Caputo-Erdélyi-Kober type fractional differential system. Define $f(t,x) = \frac{5}{3^{\frac{11}{6}}}c^{-\frac{5}{4}}x^{\frac{2}{3}}\ln(t+1)$, $I_1(x) = \frac{1}{10}c^{-\frac{1}{2}}x$. Then $d_1 = L_1 = \frac{1}{10c^{\frac{1}{2}}}$. By direct calculation, we have $M \approx 1.15c^{\frac{1}{4}}$, $M^2 L_1 = M^2 d_1 \approx 0.132 < 1$. Choosing $\varsigma(t) = \Gamma(0.25)c^{\frac{3}{4}}t^\sigma$, $\rho = c$, a direct calculation yields:*

$$
{}^C D_{0^+}^{0.75,ct^\sigma} \varsigma(t) = 4t^{\frac{1}{4}\sigma}, \ \|\varsigma\|_{\alpha,\psi}^2 = \frac{32}{3}c, \ \left(\frac{1}{2} - \frac{M^2 d_1}{2}\right)\|\varsigma\|_{\alpha,\psi}^2 \approx 4.6c > \rho,
$$

*then*

$$\frac{\sum_{i=0}^{n}\int_{s_i}^{t_{i+1}}\sup_{x\in\Omega(\rho)}F_i(t,x(t))\psi'(t)dt}{\rho}=\frac{(\int_0^{t_1}+\int_{s_1}^1)c\sigma t^\sigma\sup_{x\in\Omega(\rho)}3^{-\frac{5}{6}}c^{-\frac{5}{4}}x^{\frac{5}{3}}\ln(t+1)dt}{c}<\frac{1}{50\sigma},$$

*and*

$$\frac{2\sum_{i=0}^{n}\int_{s_i}^{t_{i+1}}F_i(t,\varsigma(t))\psi'(t)dt}{\|\varsigma\|_{\alpha,\psi}^2-2\sum_{i=1}^{n}\int_0^{\varsigma(t_i)}I_i(s))ds}=\frac{\frac{2}{3^{\frac{5}{6}}}\Gamma^2(0.25)c(\int_0^{t_1}+\int_{s_1}^1)\sigma t^{2\sigma-1}\ln(t+1)dt}{\|\varsigma\|_{\alpha,\psi}^2-\frac{1}{10}c^{\frac{-1}{2}}(\varsigma(t_1))^2}>\frac{1}{10\sigma},$$

*so that the condition $(A_2)$ holds. From Theorem 1, for each $\lambda\in]10\sigma,50\sigma[$, the system (27) possesses at least three distinct classical solutions.*

## 4. Conclusions

In this paper, we have investigated a new class of instantaneous and non-instantaneous impulsive boundary value problem involving the generalized $\psi$-Caputo fractional derivative. Based on properties of $\psi$-Caputo-type fractional operators and the three critical points theorem, the multiplicity results have been established. This problem is novel and hasn't been touched yet. By choosing special kernel functions in the $\psi$-Caputo fractional derivative, some existing results which focus on the classical fractional operators have been improved and supplemented.

**Author Contributions:** Formal analysis, D.L. and Y.L.; Investigation, D.L. and Y.L.; Methodology, D.L.; Writing—original draft, D.L. and Y.L.; Writing—review and editing, F.C. and X.F. All authors have read and agreed to the published version of the manuscript.

**Funding:** This research was funded by National Natural Science Foundation of China grant numbers 12101481, 62103327; Young Talent Fund of Association for Science and Technology in Shaanxi, China grant number 20220529; Young Talent Fund of Association for Science and Technology in Xi'an, China grant number 095920221344.

**Data Availability Statement:** Not applicable.

**Acknowledgments:** The authors would like to thank the editor and reviewers greatly for their precious comments and suggestions.

**Conflicts of Interest:** The authors declare no conflict of interest.

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
