# Peer review of "Instantaneous and Non-Instantaneous Impulsive Boundary Value Problem Involving the Generalized ψ-Caputo Fractional Derivative"

_fractalfract, doi:10.3390/fractalfract7030206_

Round 1

Reviewer 1 Report

The introduction with a linear example with the given righthandside of instantaneous vs non-instantaneous impulses should be provided with the explanation of the differences at the simplest possible level should be provided. The paper extends the results to treat Caputo derivatives and integrals with weights, exponential functions in the example. Another example should be provided with different weights eg power functions. Reduction to the known results when no weights appear should also be included. The redaction should be improved. For example `the problem... exists solution' is not correct, rather `the problem... possesses a solution' or `for the problem ... there exists a solution' are correct versions used many times in the paper.

Round 2

Reviewer 1 Report

Now it can be accepted.